# Molecular Pathways Implicated in Radioresistance of Glioblastoma Multiforme: What Is the Role of Extracellular Vesicles?

**DOI:** 10.3390/ijms24054883

**Published:** 2023-03-02

**Authors:** Pavel Burko, Giuseppa D’Amico, Ilia Miltykh, Federica Scalia, Everly Conway de Macario, Alberto J. L. Macario, Giuseppe Giglia, Francesco Cappello, Celeste Caruso Bavisotto

**Affiliations:** 1Section of Human Anatomy, Department of Biomedicine, Neuroscience and Advanced Diagnostics (BIND), University of Palermo, 90133 Palermo, Italy; 2Department of Human Anatomy, Institute of Medicine, Penza State University, 440026 Penza, Russia; 3Department of Microbiology and Immunology, School of Medicine, University of Maryland at Baltimore-Institute of Marine and Environmental Technology (IMET), Baltimore, MD 21202, USA; 4Euro-Mediterranean Institute of Science and Technology (IEMEST), 90139 Palermo, Italy; 5Section of Human Physiology, Department of Biomedicine, Neuroscience and Advanced Diagnostics (BIND), University of Palermo, 90133 Palermo, Italy

**Keywords:** glioblastoma multiforme, radioresistance, extracellular vesicles, intercellular communication, stem cells, tumor heterogeneity, tumor microenvironment, hypoxia, metabolic reprogramming, chaperone system, non-coding RNA, DNA repair, theranostics, personalized medicine

## Abstract

Glioblastoma multiforme (GBM) is a primary brain tumor that is very aggressive, resistant to treatment, and characterized by a high degree of anaplasia and proliferation. Routine treatment includes ablative surgery, chemotherapy, and radiotherapy. However, GMB rapidly relapses and develops radioresistance. Here, we briefly review the mechanisms underpinning radioresistance and discuss research to stop it and install anti-tumor defenses. Factors that participate in radioresistance are varied and include stem cells, tumor heterogeneity, tumor microenvironment, hypoxia, metabolic reprogramming, the chaperone system, non-coding RNAs, DNA repair, and extracellular vesicles (EVs). We direct our attention toward EVs because they are emerging as promising candidates as diagnostic and prognostication tools and as the basis for developing nanodevices for delivering anti-cancer agents directly into the tumor mass. EVs are relatively easy to obtain and manipulate to endow them with the desired anti-cancer properties and to administer them using minimally invasive procedures. Thus, isolating EVs from a GBM patient, supplying them with the necessary anti-cancer agent and the capability of recognizing a specified tissue-cell target, and reinjecting them into the original donor appears, at this time, as a reachable objective of personalized medicine.

## 1. Epidemiology of Glioblastoma Multiforme

Glioblastoma multiforme (GBM) is one of the most common primary malignant brain tumors and is characterized by cells with astrocyte differentiation. According to the World Health Organization (WHO), it has an incidence between 0.59 and 3.69 per 100,000 people worldwide and accounts for over 60% of all adult brain tumors [1]. Despite being rare, because of its poor prognosis, GBM contributes to 2.5% of all cancer mortality, and the median overall survival is approximately 14–17 months [2].

As in most cancers, age is a factor contributing to GBM incidence [3,4,5], and, on average, it is diagnosed at the age of 65 years [5], with a peak age of 75–79 years [6]. The older age of diagnosis usually means a worse prognosis. Elderly GMB patients have significantly shorter survival times than younger adults [7,8].

### 1.1. Modern Treatment

Therapy for GBM patients includes surgical resection of the tumor and fractionated radiation therapy concurrent with Temozolomide chemotherapy. However, other approaches to GBM treatment have been developed, as discussed below.

#### 1.1.1. Chemotherapy

The drugs often used to treat GBM are Temozolomide, intravenous Carmustine, Carmustine wafer implants, Bevacizumab, Vorinostat, Olaparib, Lomustine, and Valproic acid. The principal features of drugs currently used in chemotherapy for GBM are reported in Table 1.

#### 1.1.2. Radiotherapy

Radiotherapy is based on the effects of ionizing radiation on tumor cells. It causes direct and indirect damage to the DNA due to the water radiolysis that results in peroxide ions and radicals. The conventional regimen (dose per fraction 1.8–2.0 Gray over 30 days) remains the therapy of choice in glioblastoma.

The hypofractionation regimen significantly delays tumor growth and rarely causes side effects [38,39]. Hypofractionation radiotherapy with Temozolomide can achieve 9–20 months of survival in elderly patients (compared to 6–8 months with standard radiotherapy). However, more research is required to adjust the fractionation regimen and increase the survival rate and quality of life of patients.

Brachytherapy uses radioactive I-125 and Ir-192 isotopes to deliver ionizing radiation directly into the tumor [40,41,42,43,44]. Ir-192 is used in high-dose brachytherapy; it is removed after a certain period [42,43]. I-125 is used primarily for low-dose brachytherapy, and its capsules often remain in the body as their radiation intensity does not cause significant side effects [41,43]. Standard treatments combined with high-dose brachytherapy between surgery and external radiotherapy have been evaluated [40]. The study showed an increase in overall survival and survival without tumor progression. Furthermore, brachytherapy for inoperable patients can significantly increase their overall survival compared to life-sustaining treatment [42,43]. The main advantage of brachytherapy is its localized action and reduced distance between the radiation source and the tumor, which can lead to a reduced rate of tumor recurrence. However, inadequately high doses cause a high rate of radionecrosis in some patients.

Radiosurgery has the most remarkable efficacy during tumor progression or tumor recurrence [45]. The average survival rate in patients with glioblastoma recurrence after radiosurgery is 9 months. There is an improved survival rate in recurrent glioblastoma patients, as well as a reduction in the side effects of radiosurgery and Bevacizumab, which limit tumor growth by inhibiting angiogenesis [46].

### 1.2. Radioresistance

The response to radiotherapy is not consistent for all patients. The high genetic and molecular variability of GBM makes it difficult to predict the patient’s response to therapy. Radioresistance in some GBMs leads to poorer outcomes following radiotherapy. Aggressive growth, early and almost inevitable recurrence, and a poor prognosis require novel studies on radioresistance to improve the survival rate and quality of life [47]. Despite extensive research on GBM resistance, its mechanisms are still poorly understood.

Replication stress (RS) is a critical mechanism of DNA damage in GBM stem cells [48]. RS is an inefficient DNA replication mode in which replication forks move slowly or terminate. RS activates specific molecular processes to stabilize replication forks and prevent DNA damage. Radioresistance is associated with artificially induced RS in GBM cells. The rate of RS in glioblastoma stem cells (GSCs) is higher, as indicated by higher levels of the following parameters: replication protein A, single-stranded DNA binding protein, and DNA damage markers [48,49].

Tyrosine kinase MET is involved in the signaling cascade of DNA damage repair under ionizing radiation and is required for proper cell migration during embryonic development [50]. It enhances cell survival, angiogenesis, invasion, and metastasis in cancer [51]. The main mechanisms induced by MET are (1) activation of AKT kinase and the subsequent downstream DNA repair effectors; and (2) phosphorylation and cytoplasmic retention of the p21 protein, which has an anti-apoptotic impact.

Radioresistance is much more than a handful of surviving cells; it is a crucial mechanism in establishing the therapy resistance of the whole tumor. Remarkably, isolated glioblastoma cell lines do not show as much resistance due to the lack of cell interactions required for the development of radioresistance [52].

## 2. Adaptation Mechanisms in Glioblastoma’s Resistance to Radiotherapy

Radiotherapy is the most effective treatment method for most primary tumors of the central nervous system. However, its efficacy is limited by the phenomenon of tolerance to radiation therapy, characterized by uninterrupted tumor growth after radiation exposure and being a risk factor for metastatic disease, which requires a change in the standard patient management protocol [53]. Radioresistance is a process in which the tumor cells or tissues adapt to the radiotherapy-induced changes and develop resistance to the radiotherapy [54]. The factors involved in this phenomenon include cancer stem cells (CSCs), the chaperone system, tumor cell plasticity and heterogeneity, microenvironment, hypoxia, metabolic reprogramming, gene regulation, microRNAs (miRNAs), DNA repair, and the cell cycle (Figure 1), which are discussed in the following subsections.

### 2.1. Glioblastoma Stem Cells

The tumor tissue consists of two types of cells: cancer stem cells (CSCs) (0.01–5%) and non-CSCs (99.9–95%). The former have the capabilities of proliferation, differentiation, and self-renewal and constitute the source of cancer persistence. The non-CSCs constitute the bulk of the tumor mass, along with the differentiated and death-committed cells [55]. The presence of CSCs in the tumor mass partially explains the phenomenon of cell resistance to ionizing radiation [56].

CSCs are a tumor cell population with properties that distinguish them from other malignant cells, namely the ability to initiate carcinogenesis, sustain tumor proliferation, differentiate into all cellular subpopulations present in the primary tumor, and engage in unlimited self-renewal [57,58].

There are two main ways to explain the origin of CSCs. One postulates their establishment from postnatal stem cells, whereas the other proposes that CSCs originate by reprogramming differentiated tumor cells [59]. In addition, epigenetic reprogramming mechanisms, like those in embryonic stem cells, also play a role in the formation of CSCs [57].

Some reports describe the existence of self-renewing tumor-forming cells in glioblastoma and other types of gliomas capable of multilinear differentiation with stem cell-typical markers, according to which they are considered GSCs [60,61,62,63,64,65]. These may be critical factors in treatment failure and poor patient outcomes [66]. These GSCs, along with other indicators, express the special marker CD133 (prominin-1) that participates in the differentiation of GSCs and their self-renewal, which has a key role in carcinogenesis [55] and in the development of resistance to radiotherapy [67]. CD133-positive cells can survive high-dose radiotherapy and favor tumor relapse, despite the concomitant damage to tumor blood vessels [68], which increases after radiation exposure [67]. CD133 antigen expression is considerably higher in regrowing glioma tissue than in primary tumor tissue obtained from recently diagnosed patients [55]. The proportion of CD133-positive cells is an independent factor important for tumor regrowth and patients’ survival [67]. CD133-positive tumor cells enable the DNA damage checkpoint in feedback to radiation and a more effective fix for radiation-induced DNA damage, which may cause, at least in part, the radioresistance of CD133-positive glioma-initiating cells (GICs) [69,70]. Additionally, they show resistance to apoptosis [71].

The proliferating cell nuclear antigen (PCNA)-associated factor (PAF) plays an essential role in GSC’s self-renewal, radioresistance, and tumorigenicity [72]. PAF is predominantly overexpressed in GSCs, controls the sliding of PCNA along the DNA, and facilitates the switch from error-free to error-prone DNA synthesis [72]. A negative correlation between PAF and overall survival was observed [72]. GSCs with high Cathepsin L co-expression also have extraordinarily low radiosensitivity [70].

### 2.2. Tumor Plasticity and Heterogeneity

Tumor heterogeneity is one of the tenets of tumor progression, metastasization, development of resistance to therapy, and recurrence [73,74].

Two types of heterogeneity, intra-tumoral and inter-tumoral, cause difficulties in managing GBM [75,76,77,78]. Heterogeneity includes various alterations at the transcriptional, methylation, and mutational levels [79]. Single-cell-derived subclones can be a source of phenotypically heterogeneous progenies [80]. In GBM, inter-tumoral heterogeneity contributes more than intra-tumoral one to overall tumor heterogeneity [81].

In GBM, tumor cells from different locations in the same tumor mass will develop different extra mutations and show diverse epigenetic or phenotypic variants [75]. Intra-tumoral heterogeneity is thought to contribute to disease progression and, at least partially, to the different responses and resistance to treatment [82,83].

A fluorescence-guided multiple sampling approach with integrated genomic analysis of GBM tissues identified the various phenotypic profiles of tumor clones present in the same malignancy and established that each fragment of the tumor includes a complicated hierarchy of the clone members [84]. Furthermore, it was shown by a single-cell RNA sequencing assay that GBM has numerous cell states with different transcriptional programs and dynamic transitions [85]. Molecular and cytogenetic analyses demonstrated that the GSCs, or typical ancestor cells, bear distinctive genetic anomalies and various tumorigenic potentials [64]. Variable stem cell or regenerative activity was reported for subclones in each GBM [86]. Due to the different reactions to genotoxic damage by GSCs, the response to radiation therapy may also differ in radioresistance.

Another critical point is that tumor-cell plasticity allows for adaptation to intra- and extracellular changes. For example, bidirectional plasticity by epigenetic reprogramming is possible via a set of neuro-developmental transcription factors, with the possibility of completely reprogramming differentiated cells of GBMs to GICs [87].

CSCs encompass two types of the hierarchical model of cerebral cell differentiation: symmetric subdivision to support a pool of CSCs and asymmetric subdivision to give rise to the various populations that form GBM. Differentiated tumor cells can reverse their directionality and modify their hierarchy to form CSCs and non-CSCs progeny [88,89].

### 2.3. Tumor Microenvironment and Hypoxia

The effectiveness of radiation therapy also depends on the microenvironment of the tumor, in addition to factors inside the body that affect radioresistance. The tumor microenvironment is the result of interactions between the tumor cells and surrounding cells and molecules and contributes to GBM tumorigenesis and regrowth [90,91]. The GBM microenvironment includes blood vessels, glioma stem cells, astrocytes, fibroblasts, neural precursor cells, extracellular and vascular pericytes, different types of non-neoplastic stromal cells, and immune cells [90,91]. It also includes signaling molecules (e.g., cytokines, chemokines, and growth factors) and the extracellular matrix, all of which generate a hypoxic, inflammatory, and immunosuppressive milieu [57,90,91]. Various biomolecules are derived from cells within the tumor mass to support its progression and growth. All these cells and molecules in the tumor microenvironment most likely participate in the radiation-induced response along with the changes in phenotype, gene expression, and functions, mechanisms that cause the release of growth factors, activation of tumor-associated fibroblasts, induction of inflammation, and hypoxia [57,92]. Thus, the cellular radioresistance of GBM depends on the tumor microenvironment. This is supported by the fact that CD133-positive cells are comparatively more radioresistant in intracerebral growth conditions than in vitro [93].

CSCs are clustered in some regions of the microenvironment called niches, which make available autocrine signaling and signals outgoing from tumor-associated fibroblasts, immune and endothelial cells, and extracellular matrix components [94,95]. Even though accurate data on the structure of niches and their signaling interaction with the tumor are scarce, it is accepted that the microenvironment supplies CSCs with oxygen and nutrients, supports their functions, and protects them against radiation [96].

The concept of a perivascular niche for GSCs was advanced in 2007 [97] and that of a periarteriolar niche was proposed in 2015 [98], which pointed to the type of vessels walled by GSCs. This distinction of the niches into “perivascular” and “periarteriolar” is essential because, in most cases, “perivascular” implies capillaries [97,99]. However, it is necessary to bear in mind that while arterioles are transport vessels, capillaries are exchange vessels [100]. This means that there is no release of oxygen from the lumen of the arterioles into the surrounding tissues, explaining the occurrence of hypoxic areas. Thus, the place for the residence of GSCs is the hypoxic periarteriolar niches. It has been established that oxygen concentration affects the reaction of mammalian cells to radiation [101]. Hypoxia is a key condition for the CSCs to maintain their stemness [102]. Oxygen is a strong radiosensitizer, and its presence is necessary for forming radiation-induced reactive oxygen species (ROS), thus contributing to cell death. Therefore, a shortage of oxygen increases radiation resistance [103,104,105]. Additionally, hypoxic niches up-regulate ROS scavenging, thus decreasing ROS levels [57,106]. Hypoxia is a cause of increased expression of VEGF and hypoxia-inducible factor (HIF)-1α, which were identified in periarteriolar niches adjacent to necrotic areas [98,107]. HIFs are significant regulators that increase the radioresistance level by activating the transcription of hypoxia response elements and activating the Hedgehog, Notch, wingless, and INT-1 (WNT) pathways. These pathways contribute to CSC maintenance [108,109]. Hypoxia mediates the functional regulation of DNA-dependent protein kinase catalytic subunit (DNA-PKcs), extracellular signal-related kinases, and HIF-1α, which causes radioresistance in GBM [110]. In turn, the increased transcription of HIF-2α provoked by hypoxia is a cause of octamer-binding transcription factor 4 (OCT-4) activation, which regulates the differentiation and self-renewal CSCs [111,112]. Another additional factor is cycling hypoxia, which means irregular and unstable perfusion of tumor tissue due to a poorly structured network of blood vessels. It leads to good and poor oxygenation periods, thereby exposing the cells to hypoxia, followed by cyclic periods of reoxygenation [113,114]. Thus, hypoxia maintains the undifferentiated state of GSCs, intensifies the colony-forming effectiveness and glioma cell migration, and activates the expression of stem cell markers [102].

### 2.4. Metabolic Reprogramming and Gene Regulation

Cancer is tightly related to metabolic disorders [115,116]. Metabolic reprogramming has been recognized as one of the ten distinctive features of tumor cells. Metabolic reprogramming is needed for both malignant transformation and tumor development, including metastasization and invasion [117]. This special type of cell-energy metabolism reprogramming is required to maintain continuing proliferation and cell growth, substituting the cellular metabolic homeostasis generally presented in normal cells [83]. It was found that several metabolic characteristics differ in almost all gliomas from normal brain tissues, including surplus production of lactate and acetate and an increase in glucose oxidation to generate macromolecular precursors and energy [118,119]. Those metabolic changes also contribute to resistance to standard treatments in GBM [120,121,122,123,124]. For instance, GBM radioresistance correlates with high rates of glycolysis and suppression of the glycolytic pathway [125,126].

The Warburg effect is seen in most tumor cells in which aerobic glycolysis occurs despite the oxygen milieu [127,128]. Tumor cells use more glucose than cells in a normal physiological state. When cancer cells uptake more glucose, the pentose phosphate pathway dominates and is a source of excess nicotinamide adenine dinucleotide phosphate (NADPH) [129]. This co-factor is a crucial player in the function of redox homeostasis and cellular antioxidant systems, protecting the cell from oxidative stress, including radiation injury [129,130]. The IDH1 gene has the most expressed regulatory NADPH-producing activity in patient-derived GBM tissue [131]. Moreover, it is the most pronounced NADPH-producing gene in GBM compared to normal brain tissue [132]. The wild-type IDH1 drives NADPH production as a reaction to radiation, contributing to radioresistance. On the contrary, the inhibition of wild-type IDH1 diminishes the NADPH level, making GBM cells radiosensitive in vivo and in vitro [132,133].

The p53 tumor-suppressor protein plays a vital role in inhibiting malignant development and cellular stress [134]. The TP53-induced glycolysis and apoptosis regulator(TIGAR) is a p53-inducible protein that defends against oxidative stress and regulates glycolysis. TIGAR can decrease ROS levels and lower sensitivity to other ROS-associated apoptotic signals and p53 [135]. Knockdown of TIGAR intensifies DNA damage by overwhelming the pentose phosphate pathway, thereby reducing the radioresistance of glioma cells to radiation exposure [136,137].

The ATPase family AAA domain-containing 3A (ATAD3A) is a mitochondrial enzyme that has a role in the interaction between mitochondria and the endoplasmic reticulum (ER) [138,139]. Endogenic expression of ATAD3A correlates with radiosensitivity in cells of GBM. Forced ATAD3A expression considerably increased radiation resistance [140].

#### GBM Radioresistance and the Chaperone System

The chaperone system (CS) includes all the molecular chaperones, co-chaperones, chaperone co-factors, and chaperone interactors and receptors of an organism [141]. The CS components are distributed throughout the body with presence and function in all cells and tissues. The canonical functions pertain to the maintenance of protein homeostasis, and, thus, the CS plays a role in metabolism by keeping all enzymes and functionally related proteins in their native conformation in the place where they are needed. This applies to normal and tumor cells; thus, in the latter, the CS may contribute to carcinogenesis, including the development of resistance to radiotherapy. Likewise, the non-canonical functions of CS, which affect many key cellular and extracellular processes, can also play a role in carcinogenesis and pro- and anti-cancer [142]. The molecular chaperones are the chief components of the CS, and the role of some of them in carcinogenesis has been investigated. For example, Hsp60, Hsp70, and Hsp90 are released by tumor cells via extracellular vesicles (EVs) [143,144,145]. In the pathogenesis of gliomas, molecular chaperones play different roles. Hsp90 and Hsp47 favor angiogenesis, and Hsp70, Hsp40, and Hsp27 assist the survival pathway, promoting cancer survivability [146,147,148,149]. Hsp90 is involved in the rewiring of the metabolism and the transcription of several of the key genes that are responsible for tumorigenesis and cancer progression. Hsp90 can control metabolic rewiring, either directly by controlling the sustainability, structure, and functional activity of several metabolic enzymes or indirectly by amending the Hsp90-dependent signaling pathways involved in the expression of some proteins implicated in metabolic networks [150]. Thus, Hsp90 contributes to the radioprotective mechanisms [151,152]. Hsp70 can protect against radiation-induced apoptosis, thereby favoring glioblastoma resistance to radiation therapy [153].

### 2.5. Non-Coding RNAs

Non-coding RNAs (ncRNAs), such as long non-coding RNAs (lncRNAs) and miRNAs, may be aberrantly expressed in many tumors, indicating potential implications for cancer pathogenesis. They can play an essential role in regulating tumor radioresistance\sensitivity, and chemoresistance by controlling cell proliferation, apoptosis, DNA damage checkpoints, and other critical signaling pathways [154,155,156,157].

#### 2.5.1. miRNAs

miRNAs are short RNA molecules that control target genes by post-transcriptional silencing; a single miRNA can influence hundreds of mRNAs and regulate the expression of many genes [158]. Abnormal levels of expression of numerous miRNAs occur in GBM tumors compared with normal brain tissue [159]. For example, overexpression of 256 miRNAs and downregulation of 95 miRNAs have been found in GBM [160]. Aberrant expression of miRNAs (miR-1 [161], miR-21 [162], miR-125a [161], miR-135b [163], miR-150 [161], miR-210 [164], miR-212 [164], and miR-425 [161]) is associated with resistance to radiation therapy.

#### 2.5.2. lncRNAs

lncRNAs participate in different cellular processes and can be implicated in the development of diseases [165], including oncogenesis [166]. Additionally, they contribute to tumor radioresistance\sensitivity by controlling signal pathways, including cell apoptosis, proliferation, and metabolism; DNA damage checkpoints; and autophagy [167]. lncRNAs can regulate radiotherapy response in three ways: by acting on miRNAs, interacting with proteins to influence the cell cycle and autophagy, and operating as transcription factors to trigger downstream signaling pathways [155]. In glioblastoma, lncRNAs play a role in the establishment of radioresistance. For instance, lncRNA HMMR-AS1 is implicated in radioresistance via upregulation of irradiation-induced phosphorylation of ATM and of the levels of DNA repair proteins like RAD51 and BMI1 [168]; lncRNA HOTAIRM1 via upregulation of mitochondrial function and ROS levels in cells of GBM by controlling the expression of TGM2 [168]; lncRNA RBPMS-AS1 via downregulation of radioresistance through the miR-301a-3p/CAMTA1 axis [169]; miR-146b-5p/HuR/lncRNA-p21 axis via upregulation of β-catenin signaling pathway [170]; lncRNA SNHG18 via upregulation of suppression of semaphorin-5A [171]; lncRNA NCK1-AS1 via upregulation of the miR-22-3p/IGF1R ceRNA pathway [172]; lncRNA XIST via upregulation of the miR-329-3p/CREB1 axis [173]; lncRNA TPTEP1 via downregulation of the P38 MAPK signaling by interacting with miR-106a-5p [174]; and lncRNA linc-RA1 via upregulation of the prevention H2Bub1/USP44 combination [175].

### 2.6. DNA Repair and Cell Cycle

CSCs are radiation-resistant and have peculiar molecular properties that defend them against radiation-induced damage. The precise mechanisms of this resistance to radiation are still not completely understood, but it is believed that they depend on an increased DNA repair potential [176]. An essential part of the cell’s response to DNA damage caused by radiation is the activation of cell cycle checkpoints, which temporarily cause it to stop correcting defects in the nucleotide sequence [177]. Increased replication after radiation therapy is an adaptive response to replication stress, which includes base damage and single- and double-strand DNA breaks (DSBs). Homologous recombination repair (HRR), non-homologous end-joining (NHEJ), and alternative NHEJ, which work as backups, are the main pathways used by cells to repair the DSBs and participate in mechanisms of radioresistance in tumor cells [178]. HRR occurs preferably in the cell cycle’s late S, G2, and M-phases when a sister chromatid is present [179]. NHEJ does not need a homologous DNA template. For this reason, it can be activated at any point in the cell cycle, but it is the predominant repair pathway in G1 and G2, even when both repair pathways are working [180,181]. Maximal radioresistance is observed in the late S-phase and is explained by the increased replication level, which contributes to the process of homologous recombination [181]. Histone deacetylase (HDAC)-4 and -6 contribute to radiation tolerance in GBM by inducing DSB repair [182]. HDAC increases NHEJ editing efficiency due to the considerable HDAC inhibitor-mediated increase in Cas9 and sgRNA expression [183]. Moreover, hyperexpression of epidermal growth factor receptor (EGFR) and EGFRvIII causes radioresistance in GBM by activating both HRR and NHEJ. EGFRvIII promotes the activation of a key enzyme, DNA-PKcs, implicated in the repair of DSBs [184,185]. BMI 1 (a core component of the polycomb repressive complex 1) pairs with DNA DSB response and NHEJ in cells of GBM, which contributes to the radioresistance of GBM by recruiting DNA damage repair machinery [186].

Summing up all the pathways described above that contribute to the support and development of GBM radioresistance (summarized in Table 2), we can conclude that the diversity of the factors underlying this phenomenon requires a multipronged approach for elucidating the mechanisms involved.

## 3. Role of Extracellular Vesicles in Resistance to Radiation Therapy

### 3.1. Extracellular Vesicles

A continuous exchange of information involving molecules such as lipids, proteins, carbohydrates, and nucleic acids occurs in the human body. These molecules move to their destination in EVs, which are small vesicles coated with a phospholipid bilayer and a cargo of bioactive molecules that represents the contents of the cell in which the vesicle originated [194,195,196]. EVs are released into the extracellular space by all cell types and, consequently, are ubiquitously present in biological fluids, for example, blood [197], urine [198], saliva [199], cerebrospinal fluid [200], and breast milk [201]. EV biogenesis represents an important evolutionary advancement because the cargo is protected from degradation by ribonucleases, deoxyribonucleases, and proteases present in the extracellular space. These enzymes cannot traverse the EV’s lipid bilayer.

Based on their size, density, and mechanism of biogenesis, EVs can be sorted into three main types: exosomes, microvesicles, and apoptotic bodies [202]. Based on their size, EVs can be distinguished into small (diameter < 100 nm), medium (diameter 100–200 nm), and large (diameter > 200 nm) [203]. The main characteristics that distinguish the different subtypes of EVs are summarized in Table 3.

Today, the International Society for Extracellular Vesicles encourages the use of the term “extracellular vesicles” as a generic term for all secreted vesicles, considering the lack of consensus for the identification of specific markers to distinguish between the different subtypes of EVs [203].

### 3.2. Neuron-Derived vs. GBM-Derived EVs

There is an increasing interest in studying EVs because they are involved in communication among cells in normal physiological and pathological processes [217]. EVs and their content play an important role in tumor initiation, progression, and diagnosis [218]. GBM-derived EVs are involved in tumorigenesis, tumor microenvironment formation, angiogenesis, immune response, invasion, metastasization, and chemotherapy resistance [143,219,220,221,222].

#### 3.2.1. Role of CNS-Derived EVs in Physiological Processes

In normal conditions, EVs play an important role in sustaining diverse physiological processes, such as cell growth, development, differentiation, and apoptosis, through the interchange of genetic information and biomolecules in cell-to-cell communications [223]. In the brain, the EVs are released by neurons and different types of glial cells. Under physiological conditions, EVs transport molecules between the neurons and the glia, with consequent involvement in synaptic activity, neuronal plasticity, maintenance of myelination, and neurovascular integrity.

EVs have a substantial impact on neural development and genetic variety because of their ability to transfer various cargoes, such as protein and lipid components, signaling molecules, transcription factors, and DNA and RNA. Another potential role for EVs in developing the central nervous system (CNS) is the regulation of myelin membrane formation: the formation of the myelin membrane is downregulated by EVs released from oligodendrocytes [224].

EVs can also cross the blood–brain barrier (BBB), adding a communication channel through which systemic inflammation can modulate physiological processes in the CNS. For example, after neuronal injury, astroglial and microglial cells are activated and release exosomes that contain misfolded and inflammatory proteins and miRNAs involved in a neuroinflammatory response that affects the vitality of neurons [224]. The neuroinflammatory response can reach the periphery through the passage of exosomes through the BBB. These peripheral exosomes can be used as biomarkers for the pathogenesis of neuroinflammation and neurodegenerative disorders [224]. They act as bidirectional vehicles in brain-periphery communication, especially in neuroinflammation and aging.

EVs also play a neuroprotective role and promote neuronal regeneration in the event of injury [225,226]. EVs derived from oligodendrocytes and microglia can increase neuronal firing [227]. EVs released by neurons during neuronal remodeling are involved in synapse elimination and stimulate microglial phagocytosis. The first line of defense against pathogens in the CNS is microglia. These cells are one of the protagonists of the immune response as they express immune receptors such as toll-like receptors and produce soluble factors such as cytokines, chemokines, free radicals, and reactive oxygen species, which mediate the inflammatory response [227]. Microglia-derived EVs regulate synaptic transmission by promoting the neuronal production of ceramide and sphingosine to enhance excitatory neurotransmission [224]. Excitatory neurotransmitters, e.g., glutamate, increase the release of small EVs from neurons, oligodendrocytes, and microglia and are associated with an increase in intracellular calcium levels [228,229].

#### 3.2.2. Role of EVs in Cancer

The composition of EVs differs between healthy and cancer cells because the content of the EVs reflects the state of the secreting cell, and oncogenic processes increase the release of EVs [207]. In tumors, EVs play a key role because they can determine the fate of adjacent cells, leading to the formation of an environment that favors tumor growth [187]. Tumor-derived EVs are carriers of oncogenic factors involved in the development of GBM, and they are also responsible for their ability to infiltrate healthy brain parenchyma, which starts the formation of satellite tumors [187,220]. The vesicles produced by GBM cause suppression of the immune response against the tumor and favor the formation of new blood vessels to feed the tumor mass and the invasion of malignant cells [230]. Furthermore, GBM-derived EVs affect M2 macrophage polarization under hypoxia, thus promoting the formation of an immunosuppressive microenvironment [230].

Simultaneous injection of EVs isolated from the serum of patients with GBM and normal epithelial cells in mice caused the formation of gliomas in these mice, which confirmed the EVs’ involvement in GBM tumorigenesis [231]. The identification of transforming growth factor (TGF)-β1 in EVs isolated from the serum of patients with high-grade glioma supported the hypothesis of the involvement of GBM-derived EVs in the systemic immune response [232]. Conversely, TGF-β1 was not detected in EVs from healthy controls. TGF-β1 has pleiotropic effects, including the stimulation and activation of T cells and monocytes, but in neoplasms, the effect is mainly immunosuppressive [232].

In this regard, EVs from human GBM cell lines were studied. They carried immunosuppressive markers, including CD39, CD73, FasL, CTLA-4, and TRAIL [233]. Co-culture experiments with NK cells, CD4^+^T cells, and CD8^+^ cells revealed a downregulation of the activation state, reduced cytokine production, and increased apoptosis of CD8^+^T cells. Other upregulated markers in this population were CD39, PD-1, and EGFR [233]. It was found that 90% of all GBM patients showed aberrant expression of at least one of the following EV-markers: EGFR, EGRRvIII, podoplanin, and IDH1 [234]. Other EV components, such as mRNA and miRNA, also have the potential as tumor diagnostic markers. miRNAs can be exchanged between cells via exosomes and their detection and analysis provides information about the parental cell [143,223,235,236,237]. The diversity of transcriptomic profiles observed in glioma cells is mirrored in EVs derived from these cells. The expression levels of one small non-coding RNA (RNU6-1) and two miRNAs (miR-320 and miR-574-3p) are useful parameters for diagnosing GBM [238]. Exosomal miR-21 is also a useful marker for the diagnosis and assessing the prognosis of GMB because its levels are correlated with tumor recurrence and metastasis [239].

### 3.3. Bidirectional Communication between GBM and the Surrounding Tumour Microenvironment 

The surrounding tumor microenvironment (TME) in glioblastoma is highly heterogeneous. It consists of cancerous and non-cancerous cells, including endothelial cells (ECs), immune cells, glioma stem cells (GSCs), and astrocytes, as well as non-cellular components, such as the extracellular matrix [240]. TME is considered a crucial supporter of GBM progression, and EVs have recently been identified as an essential means of bidirectional communication between tumors and TME [241].

Rapidly growing GBM is accompanied by the formation of hypoxic areas [242]. Lack of oxygen is the cause of the formation of new blood vessels to supply oxygen and nutrients to the tumor. GBM-derived EVs have been implicated in vascular endothelial cell proliferation, migration, and tubulogenesis by releasing angiogenic proteins [243]. Mainly released by hypoxic GBM cells, vascular endothelial growth factor (VEGF)-A promotes the proliferation and migration of ECs toward hypoxic regions of GBM [244]. The result of neoangiogenesis in GBM is a highly disorganized and leaky network of vessels within areas of extreme chronic hypoxia. GBM-derived EVs, grown under hypoxic conditions, alter the phenotype of ECs to induce angiogenesis ex vivo and in vitro [245,246].

Other intercellular communication alterations occur in astrocytes, the most abundant glial cells, representing about 50% of the volume of the human brain [247]. EV-mediated crosstalk between glioblastoma cells and astrocytes supports tumor growth. Consequently, GBM-derived EVs have been implicated in altering the phenotype of normal astrocytes. Normal astrocytes exposed to GBM-derived EVs produce a tumor growth-stimulating secretome that includes VEGF; epidermal-growth, fibroblast-growth, and colony-stimulating factors; and Interleukins 10 and 19 (IL-10 and IL-19). GBM-derived EVs can be involved in the remodeling of astrocytic projections and disruption of the BBB in patients, favoring tumor invasiveness [248]. Astrocytic end feet are directly involved in the structure of the BBB. They are displaced during the development of the GBM, causing the loss of astrocyte-vascular coupling and the formation of openings in the BBB [249]. In addition, the remarkable proliferation of ECs in the GBM areas with increased hypoxia, disrupts tight junctions, leading to the loss of integrity of the BBB [250].

GBM initiation and growth are attributed to its ability to evade the immune response. EVs derived from GBM cells regulate the immune response to tumor growth via PD-L1/PD1 signaling [251]. PD-L1 associated with the EVs can directly bind the PD1 receptor on the surface of infiltrating T cells in the brain, inhibiting their activation and consequently promoting immunosuppression [251].

### 3.4. Role of EVs in Tolerance to Radiation Therapy

Therapeutic resistance remains a major obstacle to successful cancer treatment. Multiple mechanisms of resistance to therapy mediated by EVs have been described for various tumors, including breast, prostate, lung, kidney, ovarian, hematological, pancreatic, stomach, and brain cancers [252]. Diverse resistance mechanisms have been discovered in which EVs are involved. For example, EVs derived from resistant tumor cells and tumor support cells transfer the genomic and proteomic cargo (mRNA, miRNA, lncRNA, spliceosomes, and proteins) to the glioma treatment-sensitive cells, which improves their acquisition of a resistant phenotype and, by doing so, facilitates chemo- and radioresistance in GBM [187,188,189,190,191,192]. Transferring transcripts of DNA repair enzymes, such as alkylpurine-DNA-N-glycosylase and MGMT, results in increased DNA repair capacity in recipient cells [193]. GSCs-derived EVs enhance radiation resistance in GBM [191]. The regulation of DNA repair pathways and the CSCs’ state are coordinated by EV-mediated secretion of miR-603, leading to acquired radioresistance and cross-resistance to DNA alkylating agents and producing the treatment-resistant CSC phenotype [191]. EVs have an impact on the biological properties of GSCs, such as cell viability, invasion, and radioresistance. In this regard, the contribution of the hypoxia-inducible factor-1α (AHIF), transported by EVs, to the upregulation of radioresistant GBM cells was studied [192]. It was found that the expression of AHIF is highly represented in GBMs in response to radiation therapy, and suppression of AHIF in GBMs decreases cell radioresistance. Furthermore, EVs derived from AHIF-knockdown cells inhibited GBM radioresistance.

## 4. Conclusions and Future Perspectives

GBM remains cancer with a high mortality rate, notwithstanding numerous research efforts and clinical trials using a variety of drugs and radiation. Despite the technological progress that has improved medical equipment and methods of radiation therapy, patient survival is still low. The development of resistance to radiation therapy by tumor cells is a frequent obstacle to therapy. Some progress has been made in the understanding of radioresistance mechanisms, as discussed in this brief review, but much needs to be elucidated at the molecular level to facilitate the development of efficacious treatments.

EVs are involved in different ways in the onset and rapid growth of GBM. Tumor-derived EVs are oncogenic factor carriers involved in the initiation, progression, and formation of a resistant phenotype in GBM. Additionally, EVs reflect the transcriptomic profiles of the GBM cells that secret them. Therefore, EVs offer a means for diagnosis, prognostication, patient monitoring, and treatment, representing a promising theranostics tool. EVs can be used to deliver drugs directly to the tumor. In the literature, attention is directed to therapeutic agents such as radiolabeled compounds, quantum dots, plasmonic nanobubbles, liposomes, magnetic nanoparticles, polymer-conjugates, and nanovesicles, dendrimers linked with targeting agents or antitumor molecules and imaging substances [253,254,255,256,257,258,259,260,261,262]. However, there is still a need for useful tools, particularly regarding tumors of the nervous system, and EVs provide an alternative. One crucial point is that EVs can overcome the BBB. Consequently, efforts have been dedicated to developing nanomaterials, including EVs, that can penetrate the BBB [263,264]. The nanomaterials used in the treatment of GBM must meet several criteria. For example, the EVs with the active molecules must be: (1) exclusively released by the tumor cells; (2) produced by viable cells in the tumor mass rather than only loose cells undergoing necrosis or apoptosis; (3) represent the biomolecular diversity, i.e., heterogeneity, of the entire tumor; (4) penetrate through the BBB; (5) effective for specific interaction with, and penetration into the diseased regions of the brain; (6) endowed with a long-term half-life and long preserved delivery capability while in circulation; (7) capable of protecting the cargo from degradation; (8) easily detectable in tissues and fluids and (9) amenable to quantification and manipulation without major technical difficulties [265,266,267,268,269,270,271]. Most of these conditions are met by EVs, making them promising theranostics tools for studying and developing efficacious, personalized GBM treatment [269,270,271].

## Figures and Tables

**Figure 1 ijms-24-04883-f001:**
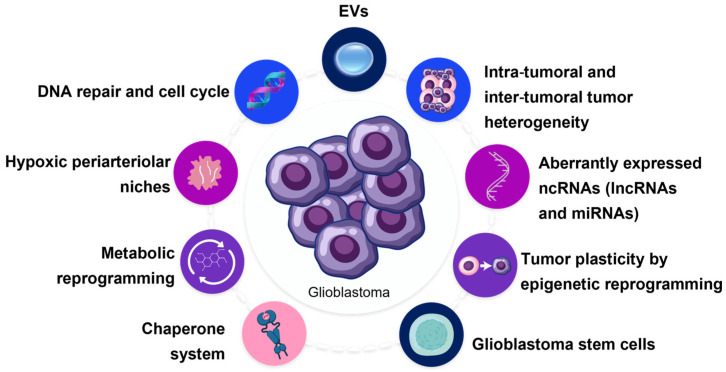
Schematic representation of factors and pathways involved in glioblastoma resistance to radiotherapy.

**Table 1 ijms-24-04883-t001:** Drugs currently used for chemotherapy of GBM.

Drug	Target	Biological Effects	Advantages	Limitations/Concerns	References
Temozolomide	Alkylation or methylation of guanine N^7^ or O^6^ and adenine N^3^	Induction of guanine binding to thymine instead of cytosine, leading to extensive DNA damage and, eventually, apoptosis	Rapid and complete absorption. Weak plasma protein binding. Blood-brain barrier permeability. Use in patients with kidney and liver malfunction	Myelosuppression and lymphopenia. Downregulation of the O^6^-methylguanine-DNA. methyltransferase gene (MGMT). Lymphoblastic leukemia	[9,10,11,12,13,14]
Carmustine	DNA and RNA alkylating agent	Binds to and modifies glutathione reductase, which leads to cell death in tumor cells	Recurrent GBM	Pulmonary fibrosis, bone marrow suppression, optical toxicity	[15,16]
Carmustine in biodegradable polymer			Placement of wafers directly into the resection area allows more effective local treatment, resulting in improved outcomes and reduced toxicity	Cerebral oedema, intracranial hypertension, infections, seizures, and thromboembolic events	[17,18,19,20]
Bevacizumab	Vascular endothelial growth factor (VEGF)	Inhibition of VEGF	Inhibition of Vascular Endothelial Growth Factor A leads to a decrease in the growth of new blood vessels, reducing the vascularization of GBM. Increases recurrence-free period in recurrent GBM	Pulmonary embolism, arterial hypertension, and hematologic toxic effects	[21,22,23,24,25]
Vorinostat	Inhibitor of histone deacetylases 1, 2, 3, and 6	Inhibition of tumor growth	Inhibition of growth of tumor cells resistant to alkylating drugs. A combination of Vorinostat and Temozolomide inhibits glioblastoma growth in experimental mice	Stimulation of autophagy and inhibition of tumor cells apoptosis	[26,27,28]
Olaparib	Poly (ADP-ribose) polymerase (PARP) inhibitor	Enhance drug delivery to tumor	Higher survival rates and no damage to healthy tissues in combination with Temozolomide, and radiotherapy	Poor brain penetration	[29]
Lomustine	Alkylating agent	Formation of O^6^-chloroethyl-guanine, can be reverted by O^6^-methyl-guanine DNA methyltransferase (MGMT)	Recurrent GBM	Restricted to patients with MGMT promoter-methylated tumors. Thrombocytopenia	[30,31,32]
Valproic acid	Short-chain fatty acid	Inhibition of histone deacetylase	Chemical and metabolic stability. Increasing tumor cell sensitivity to ionizing radiation	Thrombocytopenia, fatigue, and hypertension	[33,34,35,36,37]

**Table 2 ijms-24-04883-t002:** Factors and pathways involved in GMB resistance to radiation therapy.

Factor or Pathway	Properties/Mechanism	Reference
EVs	Transfer the genomic and proteomic cargo (mRNA, miRNA, lncRNA, spliceosomes, and proteins).	[187,188,189,190,191,192]
	Transfer the transcripts of DNA repair enzymes.	[193]
Glioblastoma stem cells	Ability to initiate carcinogenesis, sustain tumor proliferation, differentiate into all cellular subpopulations of the primary tumor, and unlimited self-renewal.	[55]
	Expression of a particular marker CD133 (prominin-1).	[55]
	Cathepsin L co-expression.	[70]
	PAF Overexpression.	[72]
Intra-tumoral and inter-tumoral tumor heterogeneity	Heterogeneity at the transcriptional, methylation, and mutational levels.	[79]
Tumor plasticity	Epigenetic reprogramming.	[87]
Hypoxic periarteriolar niches	Reduction of ROS formation and up-regulation of ROS scavenging.	[57,106]
	Increased expression of the VEGF and HIF-1α.	[98,107]
	Activation of Hedgehog pathway, Notch, wingless, and INT-1 (WNT).	[108,109]
	Mediates the functional regulation of DNA-PKcs and ERKs.	[110]
	Activation of the OCT-4.	[111,112]
	Cyclic periods of hypoxia.	[113,114]
Metabolic reprogramming	Surplus production of lactate, acetate, and increase of glucose oxidation to generate macromolecular precursors and energy.	[118,119]
	Overexpression of the heat shock proteins Hsp27, Hsp40, Hsp47, Hsp70, and Hsp90.	[146,147,148,149,150]
	Warburg effect.	[127,128]
	Activation of a pathway that is the source of excess NADPH with extra promotion by the IDH1 gene.	[129,132]
	Activation of TIGAR.	[135,136]
	Activation of ATAD3A.	[140]
Aberrantly expressed ncRNAs (lncRNAs and miRNAs)	Control of cell cycle, apoptosis, DNA damage checkpoints, and other critical signaling paths.	[141,142,154,155,156,157]
DNA repair and cell cycle	Cells’ enlarged DNA repair potential	[175]
	Use of HRR, NHEJ, and alternative NHEJ as main pathways for DSBs processing.	[178]

**Table 3 ijms-24-04883-t003:** Main characteristics of EVs.

EVs Subtype	Size (nm)	Biogenesis Mechanism	Molecular Composition	Reference
Exosomes Small/medium EVs	40–120	Endocytic origin	ALIX, TSG101, GTPase, annexins, flotillin, and tetraspanin proteins (CD9, CD63, CD81).	[204,205,206,207,208,209,210,211,212]
Microvesicles Medium/large EVs	100–1000	Outward budding and fission of the plasma membrane	mRNA, non-coding RNAs. Selectins, integrins. CD40. ARF6.	[203,213,214,215]
Apoptotic bodies large EVs	>1000	Cell fragmentation during apoptotic cell death	VAMP3. Cytoplasmic and membrane proteins, amino-phospholipids, phosphatidylserine, and ethanolamine.	[216]

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
