# Peer review of "Molecular Pathways Implicated in Radioresistance of Glioblastoma Multiforme: What Is the Role of Extracellular Vesicles?"

_ijms, 2023, doi:10.3390/ijms24054883_

Round 1
Reviewer 1 Report
In this review, Pavel Burko et al revised the literature on the mechanisms subtended to radio resistance in glioblastoma (GBM) brain tumours. Moreover, they focused on the role of EVs in both in controlling GBM aggressiveness and as a tool to treat GBM by counteracting radioresistance mechanisms.
The review in well written and exhaustive in describing the factors that participate in radioresistance and in the compendium about EVs role and possible application.
However, the paragraph describing the chemotherapy applied to GBM patients and the drugs used to treat GBM is excessively simplified. The drugs used to treat GBM at the pharmacological level are much more than the here described ( i.e carmustine wafers or antiangiogenic compounds). Summarizing is worthy but should be exhaustive even if it is not the main topic of these review.
I recommend implementing this paragraph and the table 1.
Author Response
Dear Editor,
Thank you very much for your interest in our manuscript. We also thank the Reviewers for their very useful comments and suggestions, which we have followed in preparing the revised manuscript now submitted. We added explanations and changed various parts of the text, as indicated by the Reviewers. In addition, we made several minor changes to improve the manuscript. All new or modified parts are visible in Track Changes.
Below this message, you will find a point-by-point response to the Reviewers’ comments. We hope that this revised version of our paper is now satisfactory and meet the requirements for publication in the Special Issue "New Challenges and Opportunities: Extracellular Vesicles in Biological and Biochemical Processes" of International Journal of Molecular Sciences.
Sincerely,
Celeste Caruso Bavisotto
Author’s reply #1:
We thank the Reviewer. We really appreciate all her/his valuable and supportive comments and suggestions. As suggested, we modified the paragraph “1.1.1. Chemotherapy” and Table 1 (lines 71-78).
"Please see the attachment."

Reviewer 2 Report
This review aims to discuss the mechanisms underpinning radioresistance, the role of extracellular vesicles in resistance to radiation therapy, and the mechanisms of adaptation in glioblastoma’s resistance to radiotherapy. The theme of this review is well-chosen and its relevance is high.
However, the information is not well organized and discussed.
For example, Section 2 is difficult to read. Firstly, the subsections are each one huge paragraph, and therefore the “storyline” is difficult to follow. Please partition the text into paragraphs. Secondly, sometimes the phrases seem just thrown together without meaning. Please reformulate the subsections (especially section 2.3.) for better understanding. Given the title of the section, I understand that the focus should be on EVs, not a jumble of GBM progression mechanisms.
In contrast, section 3 is written better and presents the different mechanisms of adaptation in glioblastoma’s resistance to radiotherapy. This section is larger in size than what is written about EVs and covers several related topics.
Given the general topics discussed in this review, I think the title should be changed to reflect the contents better, as EVs are not the main focus. Also, the conclusions should be rewritten to better reflect the contents of the review. The text in the conclusions is better suited to section 2. Also, I recommend switching the order of sections 2 and 3 in the manuscript.
The main theme of the review is GBM’s resistance to radiotherapy. The authors should reformulate several sections of the manuscript in order to highlight this. As is it now, the manuscript reads more like a lot of literature thrown together and loosely linked.
On a positive note, the tables are clear and easy to read, the phrases are logical and the English is very good.
Below, I have also made some minor suggestions to improve the manuscript:
Please check the entire manuscript and write the correct GBM abbreviation.
Row 157. Correct “metasization”
Row 295. Correct “metastasization”
Row 435. Correct “metastasization”
Author Response
Dear Editor,
Thank you very much for your interest in our manuscript. We also thank the Reviewers for their very useful comments and suggestions, which we have followed in preparing the revised manuscript now submitted. We added explanations and changed various parts of the text, as indicated by the Reviewers. In addition, we made several minor changes to improve the manuscript. All new or modified parts are visible in Track Changes.
Below this message, you will find a point-by-point response to the Reviewers’ comments. We hope that this revised version of our paper is now satisfactory and meet the requirements for publication in the Special Issue "New Challenges and Opportunities: Extracellular Vesicles in Biological and Biochemical Processes" of International Journal of Molecular Sciences.
Sincerely,
Celeste Caruso Bavisotto
Reviewer 2, Comment #1:
This review aims to discuss the mechanisms underpinning radioresistance, the role of extracellular vesicles in resistance to radiation therapy, and the mechanisms of adaptation in glioblastoma’s resistance to radiotherapy. The theme of this review is well-chosen and its relevance is high.
However, the information is not well organized and discussed.
For example, Section 2 is difficult to read. Firstly, the subsections are each one huge paragraph, and therefore the “storyline” is difficult to follow. Please partition the text into paragraphs. Secondly, sometimes the phrases seem just thrown together without meaning. Please reformulate the subsections (especially section 2.3.) for better understanding. Given the title of the section, I understand that the focus should be on EVs, not a jumble of GBM progression mechanisms.
Author’s reply #1:
We thank the Reviewer for the feedback and suggestions. Please note that you can assess these edits in section 3 of the editable article, not 2, since in connection with the fulfillment of the below requirement about changing the order of sections 2 and 3.
As recommended, we have divided section 3.2 (previously section 2.2) into subsections, specifically:
- 3.2.1 Role of CNS-derived EVs in physiological processes (lines 440-482)
- 3.2.2 Role of EVs in cancer (lines 483-521).
To simplify the reading of section 3 (previously section 2) and more logically build the «storyline» of the narrative, a division into paragraphs was undertaken in accordance with the points of the topic under discussion. During re-reading, sentences with no meaning were removed (e.g., line 451-452; 467-471; 538-539; 548-550). We have also reworded the title of subsection 3.3 (previously 2.3 section) to make it more relevant to the topic covered in the section.
Comment #2:
In contrast, section 3 is written better and presents the different mechanisms of adaptation in glioblastoma’s resistance to radiotherapy. This section is larger in size than what is written about EVs and covers several related topics.
Given the general topics discussed in this review, I think the title should be changed to reflect the contents better, as EVs are not the main focus.
Author’s reply #2:
The suggestion to modify the name is a relevant remark. We have reworded it.
Comment #3
Also, the conclusions should be rewritten to better reflect the contents of the review. The text in the conclusions is better suited to section 2. Also, I recommend switching the order of sections 2 and 3 in the manuscript.
Author’s reply#3:
The order of sections 2 and 3 has been reversed. This remark is logical and justified in accordance with the concept of narration. The conclusion was modified.
Comment #4
The main theme of the review is GBM’s resistance to radiotherapy. The authors should reformulate several sections of the manuscript in order to highlight this. As is it now, the manuscript reads more like a lot of literature thrown together and loosely linked.
Author’s reply #4:
We thank the Reviewer for the accurate reading of our manuscript. We agree with the suggestion to reformulate several sections and we restructured the body of the article and rewrite some sentences to improve the article narration.
Comment #5
On a positive note, the tables are clear and easy to read, the phrases are logical and the English is very good.
Author’s reply #5:
We are sincerely glad to receive this opinion.
Comment #6
Below, I have also made some minor suggestions to improve the manuscript:
Please check the entire manuscript and write the correct GBM abbreviation.
Row 157. Correct “metasization”
Row 295. Correct “metastasization”
Row 435. Correct “metastasization”
Author’s reply #6:
We checked GBM abbreviation throughout the manuscript as requested.
Metastasization is a correct term, grammatically matching:
“…tumor progression, development of resistance to therapy, and recurrence.” Lines 189 and 190; “…invasion.” Line 277; and “…tumorigenesis, tumor microenvironment formation, angiogenesis, immune response, invasion, and chemotherapy resistance…” Lines 437-440.
However, if the Reviewer considers that there is a better option, we would be happy to replace “metastasization” with it. Please, let us know.
"Please see the attachment."

Round 2
Reviewer 2 Report
The authors have made the suggested modifications.